# Agavins Impact on Gastrointestinal Tolerability-Related Symptoms during a Five-Week Dose-Escalation Intervention in Lean and Obese Mexican Adults: Exploratory Randomized Clinical Trial

**DOI:** 10.3390/foods11050670

**Published:** 2022-02-24

**Authors:** María Blanca Silva-Adame, Arlen Martínez-Alvarado, Víctor Armando Martínez-Silva, Virginia Samaniego-Méndez, Mercedes G. López

**Affiliations:** 1Departamento de Biotecnología y Bioquímica, Centro de Investigación y de Estudios Avanzados del Instituto Politécnico Nacional, Unidad Irapuato, Guanajuato 36824, Mexico; blanca.silva@cinvestav.mx; 2Centro de Estudios Cardiometabólicos S.C., Mexico City 07760, Mexico; arlen.martinez@cetomed.com (A.M.-A.); masv_hangeles@hotmail.com (V.A.M.-S.); virsame1030@gmail.com (V.S.-M.)

**Keywords:** agavins, functional fiber, GI tolerability, obesity, branched neo-fructans

## Abstract

Agavins are prebiotics and functional fiber that modulated the gut microbiota and metabolic status in obese mice. Here, we designed a placebo-controlled, double-blind, exploratory study to assess fluctuations in gastrointestinal (GI) tolerability-related symptoms to increasing doses of agavins in 38 lean and obese Mexican adults for five weeks and their impact on subjective appetite, satiety, metabolic markers, and body composition. All GI symptoms showed higher scores than placebo at almost every dose for both lean and obese groups. Flatulence caused an intense discomfort in the lean-agavins group at 7 g/day, while obese-agavins reported a mild-to-moderate effect for all five symptoms: no significant differences among 7, 10, and 12 g/day for flatulence, bloating, and diarrhea. Ratings for any GI symptom differed between 10 and 12 g/day in neither group. The inter-group comparison demonstrated a steady trend in GI symptoms scores in obese participants not seen for lean volunteers that could improve their adherence to larger trials. Only body weight after 10 g/day reduced from baseline conditions in obese-agavins, with changes in triglycerides and very-low-density lipoproteins compared to placebo at 5 g/day, and in total cholesterol for 10 g/day. Altogether, these results would help design future trials to evaluate agavins impact on obese adults.

## 1. Introduction

Overweight and obesity are defined as “abnormal or excessive fat accumulation that may impair health” [1]. High body mass index (BMI) increases rapidly, and diseases such as musculoskeletal disorders, disability, cardiovascular disease, diabetes, chronic kidney disease, and cancers have been largely related to this risk factor in many countries [2]. A recent international nutrition report showed the alarming growing prevalence of adult obesity, with an increase from 11.8% of the global population in 2012 to 13.1% in 2016 that is not on course to change this trend by 2025 [3]; in other words, more than one-third of the world’s population is now classified as overweight or obese, a disease that affects almost all physiological functions of the body and represents a public health threat [4]. Many factors have been associated with the development and predisposition to obesity, including genetics, epigenetics, metagenomics, endocrine disruption, but there are also inter-individual factors contributing to the progression of obesity, such as the adoption of a fat- and energy-dense diet, a sedentary lifestyle and gut microbiota composition [4,5]. The gut microbiota is regarded as a contributing factor to the development of obesity and related metabolic disorders [6]; disturbance of the host-microbiome symbiosis leads to an increase in obesity and other immune-mediated pathologies; furthermore, consumption of a diet different to the one under which the human-microbiome interrelationship evolved is considered to support this phenomenon [7]. One of the major changes to the ancestral diet is a drastic reduction in non-fermentable carbohydrate intake or microbiota-accessible carbohydrates, meaning that they are metabolically available to gut microbiota, such as dietary fiber [8]; while recommendations for dietary fiber intake in adults range from 18–38 g/day, a “fiber gap” exists worldwide due to daily ingestion below the recommended levels. Efforts to change this trend include the supplementation of fibers that can interact with intrinsic ones contained in a fiber-balanced diet, not just meeting the requirements but also acting synergistically to procure health benefits [9]. In fact, fortification of foods with non-digestible carbohydrates (extracted or synthesized), such as some prebiotics, or their use as dietary supplements represents a strategy to increase fiber consumption [10].

Agavins are prebiotics and functional fiber biosynthesized in *Agave* plants representing the most abundant water-soluble carbohydrate. Classified as neo-fructans given the presence of an internal glucose unit in the molecular structure, agavins are a polydisperse mixture of complex and highly branched molecules with β(2-1) and β(2-6) linkages [11]. Among their effects on health, agavins induced weight loss, produced a positive impact on metabolic disorders and increased concentration of short-chain fatty acids (SCFAs) in overweight mice, along with a specific modulation of gut microbiota composition and a partially restored microbial diversity [12,13]. Moreover, agavins significantly decreased levels of pro-inflammatory cytokines and lipopolysaccharides (LPS), evidence of their impact on low-grade inflammation and metabolic endotoxemia and induced a reduction in lipid droplets content in the liver of obese mice [14].

With respect to clinical evidence, 5 g/day of agavins induced an enrichment in fecal Actinobacteria while 7.5 g/day increased *Bifidobacterium* abundance [15]; both agavins doses caused a slight increase in bloating, flatulence, and rumbling but did not cause diarrhea; they improved laxation, and generated a minimal GI upset, all these effects were reported in healthy young adults [16]. In fact, gastrointestinal discomfort is commonly associated with the adaptation process due to an increment in dietary and/or functional fiber intake and to the supplementation of some prebiotics. However, more information on GI adaptation to higher doses of agavins in target patient populations is scarce. Here, we aim to assess the evolution of five GI tolerability-related symptoms’ ratings while ramping-up doses weekly, in addition, to determining the effect of agavins supplementation on subjective satiety and appetite, metabolic markers, and body composition. These pilot results would help design future studies of agavins impact on human health.

## 2. Materials and Methods

### 2.1. Participants

All participants were voluntarily recruited in March–September 2019 from Mexico City, Mexico, at “Centro de Estudios Cardiometabólicos S.C.”. The inclusion criteria considered females and males between 30–60 years old, a BMI ≥ 30 kg/m^2^, and stable body weight for at least one month prior to the study. Healthy participants were also included and enrolled in this study to compare their adaptation to increasing doses of agavins and their impact in the context of no obesity-related metabolic disorders and gut microbiota perturbations. Some of the exclusion criteria were: type 1 and type 2 diabetes; hypothyroidism; currently following a weight loss diet or physical activity regime for the same purpose; weight loss > 3 kg within three months before enrollment; use of prebiotics, probiotics, or dietary fiber supplements; long-term (and within the preceding month) use of antioxidants and polyunsaturated fatty acids supplements; concomitant use of any medication influencing appetite, weight, and metabolism; antibiotic use one week prior to the study; alcohol or substance abuse; diagnosis of neurological or psychiatric disorders; alanine aminotransferase and aspartate aminotransferase enzymes concentration > 2.5 times the highest limit value; pregnancy or lactation in women; previous intestinal or bariatric surgery; intestinal absorption disorder; inflammatory bowel disease; established cardiovascular disease; chronic use of bulk laxatives and antacids, etc. Written informed consent was obtained from all volunteers. The study was revised and approved by “Comité de Ética en Investigación de Unidad Clínica de Bioequivalencia S. de R.L. de C.V.” (reference number 002216) and was registered at ClinicalTrials.gov (NCT04555447).

### 2.2. Experimental Design

This was a double-blind, placebo-controlled, parallel-group, dose-escalation, exploratory study, where lean, healthy volunteers (BMI 18.5–24.9 kg/m^2^) and obese patients (BMI ≥ 30 kg/m^2^) were randomized to either agavins or placebo group for a five-week dose-escalation period. Our study sample size was based on power calculations that used fasting plasma glucose reduction as the reference outcome, avoiding only based calculations on reported GI tolerability evaluations. Using pilot data from our laboratory, where healthy and overweight participants reached a 6.09% reduction in this variable after four weeks of agavins supplementation, and with an estimated 10% reduction in fasting glucose concentration, at 80% power and a significance level of 0.05, 14 participants were required for each intervention group. Three participants were added to compensate for a potential 20% dropout rate [17,18]. Additionally, considering published GI tolerability trials with similar prebiotics, dietary, or functional fibers [19,20,21,22,23] and high interindividual variability in GI tolerance previously reported for agave inulin [16], an additional 8 subjects per group were considered reaching *n* = 25. Nevertheless, after months of enrollment, very few volunteers for the lean healthy group responded, and even fewer participants met the inclusion criteria. For this and financing difficulties, the research team decided to stop recruiting even if our sample size had not yet been met, aiming to obtain pilot data on GI tolerability to agavins supplementation for the design and considerations in future studies of agavins impact on health.

A research assistant not involved in the study was responsible for generating each participant’s code and randomization. Investigators and participants were blinded to treatment allocation. For this study, agavins (Preventy^®^ Inulina de Agave, IMAG S.A. de C.V., Arandas, Jalisco, Mexico) and placebo (Globe^®^ 10 Maltodextrin, Ingredion Inc., Westchester, IL, USA) were provided in ready-to-use sachets contained in identical transparent re-sealable plastic bags coded by a research assistant not involved in the research. The daily dose of each supplement was ramped-up weekly from 2.5 g/day to 12 g/day. For packaging conditions, plastic sachets could be filled with 12 g of agavins at most, so we adjusted doses to include two escalations of 2 g/day instead of only one, from 5 to 7 g/day and from 10 to 12 g/day. Thus, each participant received seven sachets of the corresponding dose at every clinic visit, starting with the baseline measurements visit. All participants were instructed to dissolve the content of each sachet in water and to take the corresponding dose preferentially following their evening meal, emphasizing keeping the same intake hour daily throughout the study. Visits were programmed every week for surveillance and to resolve any questions that could arise. To assess compliance, participants were asked to return all empty sachets at every clinic visit, as well as any missed doses. Upon enrollment, a trained dietitian met with each participant. Prior to the study beginning, all volunteers were required to record their food intakes in a 24-h diet recall at baseline and every weekly visit. We aimed to evaluate the evolution of GI symptoms in a real-life context, and thus we encouraged participants not to alter their usual diet and to maintain their physical activity regime and exercise, and to sustain regular communication with research staff.

### 2.3. Tolerability-Related Gastrointestinal Symptoms, Appetite and Satiety Subjective Assessment

Participants were asked to fill out a 10-cm visual analog scale (VAS) to evaluate five relevant GI symptoms (flatulence, bloating, borborygmi, diarrhea, and abdominal pain) and appetite-related feelings. They were instructed to rate their feelings 12 h after taking their supplement dose, every day for a week for each dose. To achieve this, every subject received 7 VAS weekly, one for each GI symptom, one for subjective appetite, and another one for satiety. Stool frequency and consistency records were not considered in this pilot study. Appetite ratings were guided by “How hungry do you feel?” clamped by “I am not hungry at all” and “I have never been hungrier; I have to eat immediately”, while satiety ratings were guided by “How satisfied do you feel?” clamped by “I feel my stomach completely empty” and “I cannot eat another bite” [18]. Data were collected by measuring the distance (in centimeters) with a ruler and registered every day in a week for each of the five doses [24].

### 2.4. Body Weight and Body Composition Determination

Starting at baseline, an 8-electrode bioelectrical impedance Mbca 514 medical body composition analyzer (Seca gmbh & Co., Hamburg, Germany) was used every week at clinic visits to measure body weight (BW), relative fat mass (RFM), absolute fat mass (AFM), fat-free mass (FFM), skeletal muscle mass (SMM), total body water (TBW), and extracellular water (EW) as part of the follow-up. Height was also measured at baseline to determine BMI.

### 2.5. Metabolic Markers Assessment

At every in-person visit, fasting blood was drawn to quantify clinically relevant metabolic markers, such as fasting plasma glucose (GLU), triglycerides (TRG), total cholesterol (CHOL-T), high-density lipoprotein cholesterol (HDL-C), low-density lipoprotein cholesterol (LDL-C), and very-low-density lipoprotein cholesterol (VLDL-C). Quantifications were performed by a registered external authorized clinical laboratory.

### 2.6. Statistical Analysis

Data are presented as mean ± standard error of the mean unless otherwise stated. At baseline, comparisons within each group (lean agavins vs. placebo; obese agavins vs. placebo) of metabolic markers quantification and body composition variables were performed by an unpaired *t*-test with Welch’s correction. The same test was used to compare every tolerability-related variable, subjective appetite, and satiety ratings between agavins and placebo groups (against control comparisons) at every dose, as well as to contrast lean-agavins and obese-agavins groups (inter-group differences). On the other hand, GI symptoms, satiety, and subjective appetite evolution assessment were carried out through comparisons between the five daily doses essayed (intra-group differences) within lean and obese groups by a repeated-measures two-way analysis of variance (RM two-way ANOVA) followed by Holm–Šidák’s multiple comparisons test. To determine agavins impact on metabolic markers, intra-group differences were analyzed by a mixed-effects model, with the Geisser–Greenhouse correction followed by a Tukey’s multiple comparisons test. Similarly, intra-group differences among body composition variables were also analyzed by a mixed-effects model followed by Holm–Šidák’s multiple comparisons test. Statistical significance was considered at *p* < 0.05, and residuals were analyzed by Shapiro–Wilk, Anderson–Darling, D’Agostino, or Kolmogorov–Smirnov tests, or inspection of QQ plots. A principal component analysis (PCA) was carried out to explore and determine the links and correlations among all variables measured in this study. All statistical analyses were performed using GraphPad Prism version 9.0.0 (GraphPad Software, San Diego, CA, USA).

## 3. Results

From 52 participants enrolled in this study, 38 completed the intervention: 16 lean participants and 22 obese patients (Figure 1). Baseline characteristics of these participants are depicted in Table 1. Moreover, all subjects were instructed to maintain their dietary and exercise habits, but before starting the dose-escalation period, dietary intakes of all participants were assessed and are presented in Table 2.

### 3.1. Tolerability-Related GI Symptoms, and Subjective Appetite and Satiety Ratings

The evolution of GI symptoms ratings and of satiety and subjective appetite during the five-week dose-escalation intervention of lean participants is represented in Figure 2. RM two-way ANOVA reported agavins dose as a highly significant source of variation (*p* < 0.001) for all GI symptoms evaluated. Flatulence was the symptom with the highest ratings and rapidly reached records corresponding to an intense effect (Figure 2a), and no statistically significant differences (*p* < 0.05) between the highest doses of 10 and 12 g/day, with average ratings of 6.63 ± 2.3 and 6.63 ± 2.2. (mean ± standard deviation), respectively which still represents a strong discomfort. However, a significant effect of the subject as a source of variation (*p* < 0.001) reflects a large interindividual difference in the perception of this symptom, while the highest ratings of 8, 9, and 10 were registered in many days for most of the doses. Several participants also rated as having no effect or mild feeling throughout this escalation.

Bloating, borborygmi, abdominal pain, and diarrhea ratings were also greatly affected by agavins dose and individual’s perception (*p* < 0.001), with no significant differences (*p* < 0.05) between 10 and 12 g/day, as seen for flatulence, and only subjective satiety at 10 g/day increased more than at 12 g/day (*p* < 0.001). On the other hand, when compared to placebo controls, lean-agavins group registered a differential evaluation of GI symptoms with every dose supplemented, except for abdominal pain at 7 g/day (Figure 2d, *p* < 0.05). Furthermore, subjective satiety in lean-agavins group was only rated with a different and higher record (*p* < 0.001) than placebo at 5 g/day, whereas agavins supplementation caused a differential appetite perception when compared to lean-placebo group at all doses, excluding 10 g/day (Figure 2g, *p* < 0.05).

In contrast to GI adaptation trends to agavins intake in lean participants, ratings for all five tolerability-related symptoms in the obese-agavins group fluctuated from mild-to-moderate effect with the highest average ratings not even reaching a score of 7 (Figure 3), which shows no strong discomfort caused by any of the symptoms herein evaluated.

Interestingly, flatulence scores at 7, 10, and 12 g/day are not significantly different among them (*p* < 0.05) in the obese-agavins group, with average scores of 5.06 ± 2.07, 5.64 ± 1.83, and 5.71 ± 2.5 (mean ± standard deviation), respectively, reflecting a moderate effect perception. In addition to these results, no statistical differences were found between 7, 10, and 12 g/day for bloating and diarrhea (Figure 3b,e; *p* < 0.05). Conversely, all five GI symptoms were found significantly (*p* < 0.05) or even high significantly (*p* < 0.001) different among obese-agavins and obese-placebo groups, excluding only flatulence and bloating at 2.5 g/day, and diarrhea at 2.5 and 5 g/day. As observed in lean participants, agavins dose and individual’s perception are highly significant sources of variation (*p* < 0.001), and interestingly no differences between the highest doses of 10 and 12 g/day were found in the obese-agavins group for all symptoms. Scores for subjective appetite are higher in obese-agavins than in the obese-placebo group at all doses (Figure 3g; *p* ≤ 0.01), and satiety perception is greater in placebo controls at 5, 10, and 12 g/day (*p* < 0.05).

GI adaptation to agavins increasing doses contrast among lean and obese participants is shown in Figure 4. The five doses of agavins supplemented impacted differently in lean and obese subjects for flatulence, as well as for subjective satiety (Figure 4a,f; *p* ≤ 0.05). Similarly, bloating exhibits a differential evolution at all doses, except for 5 g/day, and borborygmi perception is much greater in lean-agavins than in the obese-agavins group (Figure 4c; *p* < 0.001), the effect observed at small doses (2.5 and 5 g/day). Altogether, these comparisons provide evidence of a different adaptation to agavins at different doses in lean and obese adults.

### 3.2. Quantification of Metabolic Markers and Body Composition

At the end of the intervention, a PCA analysis with all measured variables showed a separation of a lean-placebo subgroup, while the rest of the participants clustered together with the lean-agavins group, thus demonstrating a clear differentiating effect when consuming this functional fiber (Figure 5a).

Moreover, the PCA scores plot indicated that variations in GI symptoms and in HDL-C are closely related to agavins dose in lean-agavins group (Figure 5b); for this group, explained variance of principal component (PC) 1 is dominated by RFM, subjective appetite, flatulence, bloating, and borborygmi, whereas in the obese-agavins group (Figure 5d) PC1 is conversely dominated by HDL-C, RFM, and abdominal pain. GI symptoms in obese participants were also closely related to agavins dose as well. Finally, no clear clustering was observed for obese-agavins nor obese-placebo groups (Figure 5c).

Quantification of GLU, TRG, CHOL-T, HDL-C, LDL-C, and VLDL-C was carried out weekly during clinic visits; results are depicted in Appendix A, for lean and obese participants, respectively, both agavins and placebo groups. Lean-agavins intra-group analysis revealed no significant differences for any metabolic marker at any of the five doses, except for a significant reduction (Tukey’s multiple comparisons test; *p* < 0.05) in HDL-C concentration between 7 and 10 g/day; also, no significant differences were found when compared to lean-placebo controls. On the other hand, no obese-agavins intra-group differences were detected, but a significant effect on TRG and VLDL-C at 5 g/day and in CHOL-T after 10 g/day was observed when compared against placebo controls (*t*-test with Welch’s correction; *p* < 0.05). Furthermore, from the seven body composition parameters followed during this five-week escalation period, the intra-group analysis only showed a significant change in body weight after 10 g/day compared to baseline conditions in the obese-agavins group (Holm–Šidák’s multiple comparisons test; *p* < 0.05) as depicted in Appendix A. No change was detected in body composition variables for lean volunteers when compared to placebo controls, nor in intra-group analysis (Appendix A).

## 4. Discussion

There is increasing interest in the impact of fiber on gastrointestinal tolerance, stool frequency and consistency, gut microbiota composition, and activity [25], with the search for good digestive health that encloses a digestive system that possesses an appropriate nutrient absorption and intestinal motility, among other characteristics [26]. For agavins, growing attention exists given their prebiotic effect evidenced in animal models, in vitro, and in clinical trials for their potential in managing obesity, obesity-related disorders, and specific modulation of gut microbiota [12,13,15,27,28,29].

Gastrointestinal side effects are unavoidable yet tolerable when consuming dietary fiber as a response to fermentation and production of intestinal gas, which could be considered as a cost to benefit from the positive impact of higher intakes of dietary fiber on health [26]. Several studies of GI tolerance and adaptation to fiber intake considered symptoms like nausea, constipation, GI rumbling, bloating, flatulence, GI cramps, diarrhea, stomach noises, distension, stomach pain, among others, in response to different doses and a wide variety of fibers (oligofructose, inulin, soluble corn fiber, resistant starch, oat bran, barley bran, pullulan, arabinoxylan oligosaccharides, etc.) in a healthy population [19,20,22,30,31,32,33]. In this randomized placebo-controlled pilot trial, we present a dose-escalation intervention evaluating five daily doses, 2.5 g, 5 g, 7 g, 10 g, and 12 g that were determined from pilot data from our lab, and their impact on flatulence, bloating, borborygmi, abdominal pain, and diarrhea, in lean and obese participants. We found that every agavins dose induced a significant difference in daily ratings for the five GI symptoms herein evaluated in lean participants compared to placebo controls, except for abdominal pain at 7 g/day. Contrarily, a previous study in healthy adults receiving 5 g/day of agavins (agave fructans), which is one of the doses we tested, reported no significant change in abdominal pain after three weeks of intervention, but a significant increase in intestinal bloating producing a mild to moderate effect compared with placebo, while others have also found an increment in bloating with the same dose in healthy young adults [16,27], corresponding to a mild effect in accordance to what we observed for agavins; however we also found a significant increment in bloating scores for obese patients consuming 5 g/day of agavins, equally to lean volunteers.

Although an ambiguous term, bloating refers to a subjective sensation related to abdominal distension, considered as one of the dominant adverse effects in relation to dietary fiber intake, along with flatulence, abdominal cramps, and diarrhea [34,35], thus a mild effect detected for a dose of 5 g/day reflects a tolerable adaptation to agavins consumption. At higher doses, previously 7.5 g/day of agave fructans led to a greater daily and weekly score of flatulence, bloating, and rumblings than the control, but with a very mild effect [16]. With a similar dose (7 g/day), we found an increment in records of all evaluated GI symptoms compared to placebo in the obese-agavins group, as well as for the lean-agavins group, only excluding abdominal pain, which registered no differences. Nevertheless, inter-group comparisons from subjects supplemented with agavins reflected a differential evolution of flatulence, bloating, and diarrhea at 7 g/day, whereas only bloating scores did not differ among groups at 5 g/day.

It has been reported that flatulence is associated with other GI symptoms when following a flatulogenic diet (26.5 g of fiber) that generated a reduction in digestive comfort, likely due to an increased gas production after colonic fermentation by gut microbiota [36]. Previously, contradictory impact on flatulence after agavins consumption has been reported; while flatulence was found to be the most intense GI symptom experienced daily at 5 and 7.5 g/day [16], others detected no significant increase in flatulence after 5 g/day compared to placebo [27], both analyzed in healthy adults during a three-week period. In lean subjects, we detected flatulence as the main GI symptom associated with agavins consumption with a significant inter-individual variation and records that even reached a moderate or severe effect. It is noteworthy that analysis of the obese-agavins group revealed no significant changes in flatulence after 7 g/day, and scores for the rest of GI symptoms also did not show variations among the higher doses of 10 and 12 g/day; furthermore, in this group, daily ratings steadily incremented up until 7 g/day. After that, no differences were detected, and even a decrement in abdominal pain records was observed.

GI tolerance is reached when unwanted symptoms related to fiber intake do not persist [30]. According to this, the present intervention with agavins at daily doses that ranged from 2.5 to 12 g/day did not exhibit a GI tolerance in lean volunteers nor in obese patients. However, a steady rise in flatulence, bloating, abdominal pain, and diarrhea ratings were observed from 2.5 to 7 g/day in the obese-agavins group, followed by no significant changes between 7, 10, and 12 g/day with scores corresponding to mild to moderate effect (a maximum of 5.71 ± 2.5 for flatulence at 12 g/day), along with inter-group comparisons evidenced a GI adaptation to agavins intake in obese patients that was not seen in lean subjects. This adaptation could positively impact the adherence and participation of obese patients to evaluate further and better the agavins impact on obesity and gut microbiota modulation in a similar group of patients.

Despite the generated and valuable information, this study presents some limitations. First, the present pilot dose-escalation evaluation is part of a larger trial that includes gut microbiota composition evaluation in lean and obese participants along with the assessment of metabolic markers from which we based our sample size and power calculations, all of this as a pilot study of agavins impact in both groups, lean and obese participants. Our original sample size was *n* = 25; however, we faced serious difficulties, particularly in the lean group, to recruit volunteers meeting the inclusion criteria as well as funding challenges. For these reasons, we stopped recruiting even if the calculated sample size had not yet been met, which could be a potential reason for the non-significant effects in some variables, such as the metabolic markers. In addition to these, stool frequency and stool consistency analysis were not included and would indeed have been key variables to complete this GI adaptation overview because of agavins consumption. Second, even if we included the use of 24-h diet recalls and that we aimed to perform this evaluation independently of other lifestyle changes, a more detailed and accurate diet, and physical activity follow-up must be included in future studies. As for gender-related biases, it is well known that the menstrual cycle phase impacts some GI symptoms in many women. For instance, in a retrospective study of healthy women, 73% of participants reported experiencing at least one GI symptom of abdominal pain, constipation, diarrhea, nausea, and vomiting before menstruation, but 31% declared having multiple symptoms during menstruation or premenstrually [37]. More recently, Judkins et al. [38] found that abdominal pain, diarrhea, constipation, indigestion, and reflux varied daily and reached their highest score on day 1 of the menstrual cycle in healthy women taking oral contraceptives and transitioned from no discomfort to mild GI discomfort. Based on these findings, it may be very difficult to separate some symptoms such as abdominal pain as an effect of menstrual cramps or as a GI symptom, even when only evaluating GI discomfort fluctuations, as previously discussed [38]. Regarding this study, our results did not differentiate GI symptoms solely attributed to agavins consumption from those caused by menstruation in the female population recruited, representing 50% of lean participants and 73% of obese patients that completed the intervention.

## 5. Conclusions

This exploratory study provided valuable information on some symptoms that may cause GI distress when agavins are consumed and even more when ramping-up daily doses up until to 12 g/day to either a target patient population or healthy controls, thus shading light to better design and propose future supplementation interventions. Lean and obese adults differentially adapted to increasing doses of agavins, with a large individual and dose dependency. Furthermore, obese patients showed an adaptation to agavins intake at 7 g/day, contrary to what was observed for lean participants. Neither group reached a GI tolerance state; flatulence was the most intense symptom for both groups. A longer and larger randomized clinical trial would assess the effect of agavins in patients such as the adult obese population.

## Figures and Tables

**Figure 1 foods-11-00670-f001:**
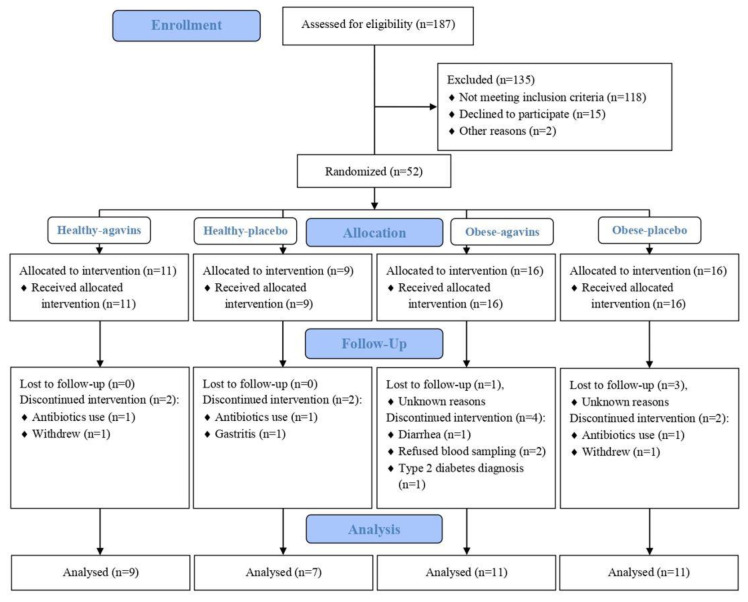
CONSORT flow diagram.

**Figure 2 foods-11-00670-f002:**
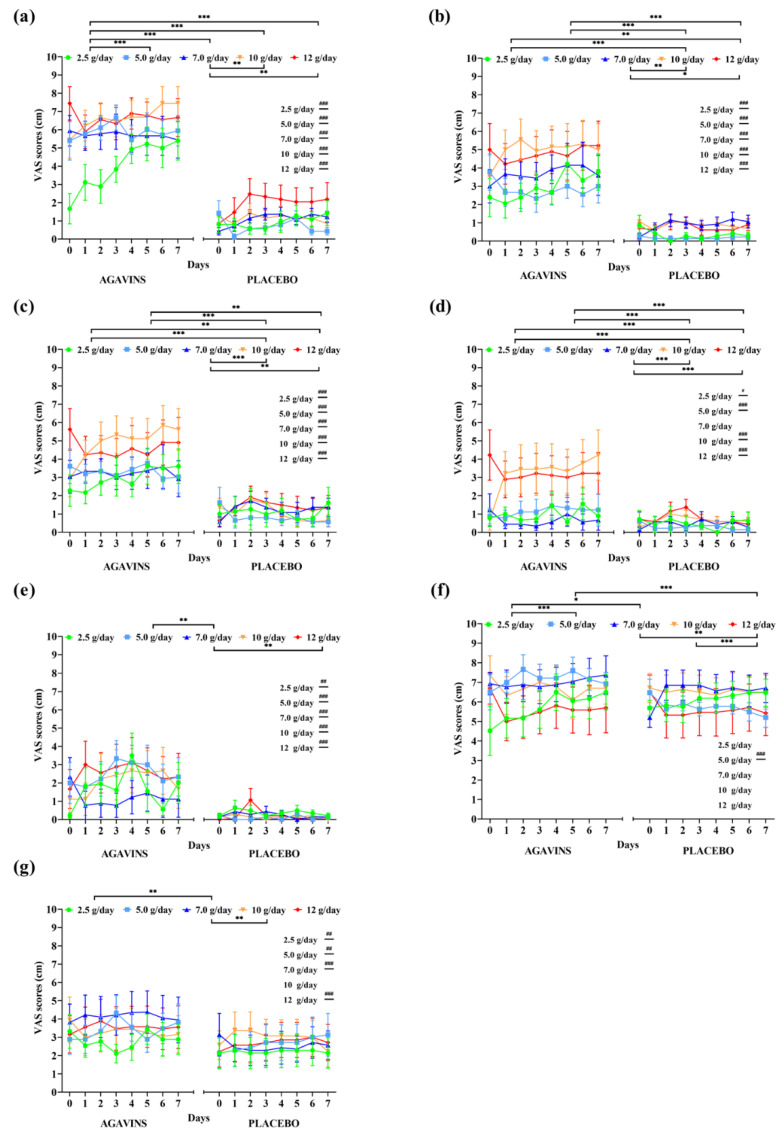
Gastrointestinal symptoms, subjective satiety, and appetite daily ratings evolution in lean individuals after intake of 2.5, 5, 7, 10, and 12 g/day of agavins or placebo during a five-week dose-escalation intervention. (**a**) Flatulence; (**b**) Bloating; (**c**) Borborygmi; (**d**) Abdominal pain; (**e**) Diarrhea; (**f**) Satiety; (**g**) Appetite. ** p* < 0.05, *** p* < 0.01, **** p* < 0.001 within group differences between doses (repeated-measures two-way ANOVA followed by Holm–Šidák’s multiple comparisons test). *# p* < 0.05, *## p* < 0.01, *### p* < 0.001 differences between agavins and placebo groups for the same dose (unpaired *t*-test with Welch’s correction). ANOVA, analysis of variance.

**Figure 3 foods-11-00670-f003:**
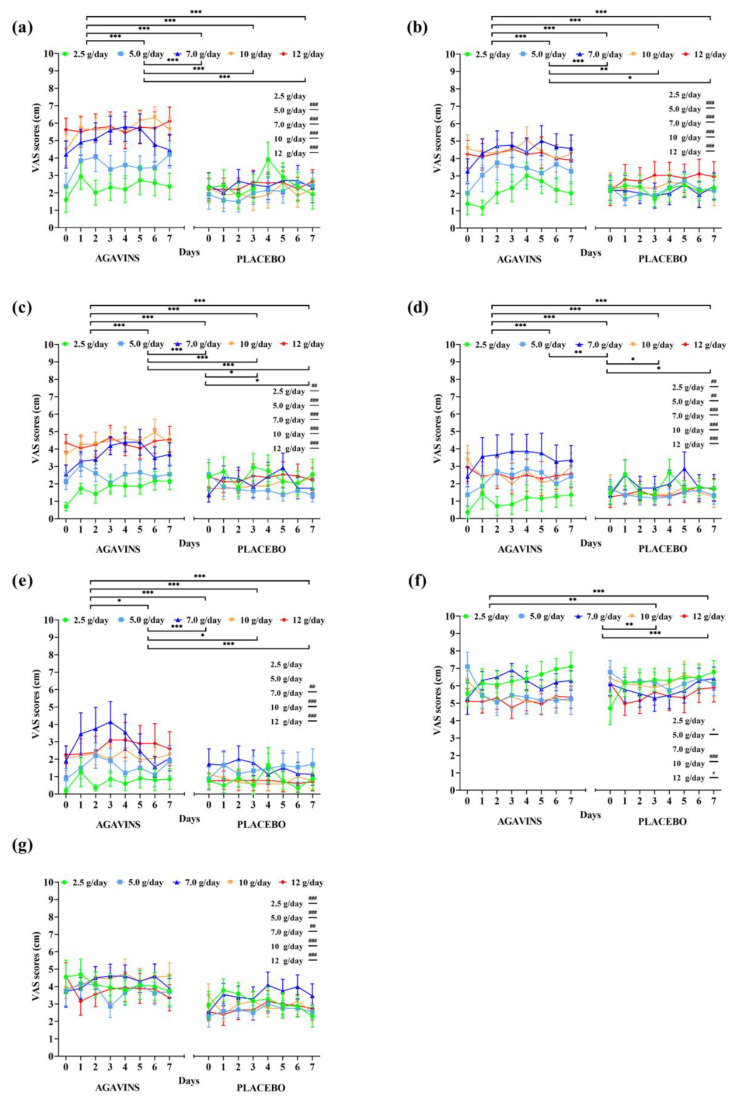
Gastrointestinal symptoms, subjective satiety, and appetite daily ratings evolution in obese adults after intake of 2.5, 5, 7, 10, and 12 g/day of agavins or placebo during a five-week dose-escalation intervention. (**a**) Flatulence; (**b**) Bloating; (**c**) Borborygmi; (**d**) Abdominal pain; (**e**) Diarrhea; (**f**) Satiety; (**g**) Appetite. ** p* < 0.05, *** p* < 0.01, **** p* < 0.001 within group differences between doses (repeated-measures two-way ANOVA followed by Holm–Šidák’s multiple comparisons test). *# p* < 0.05, *## p* < 0.01, *### p* < 0.001 differences between agavins and placebo groups for the same dose (unpaired *t*-test with Welch’s correction). ANOVA, analysis of variance.

**Figure 4 foods-11-00670-f004:**
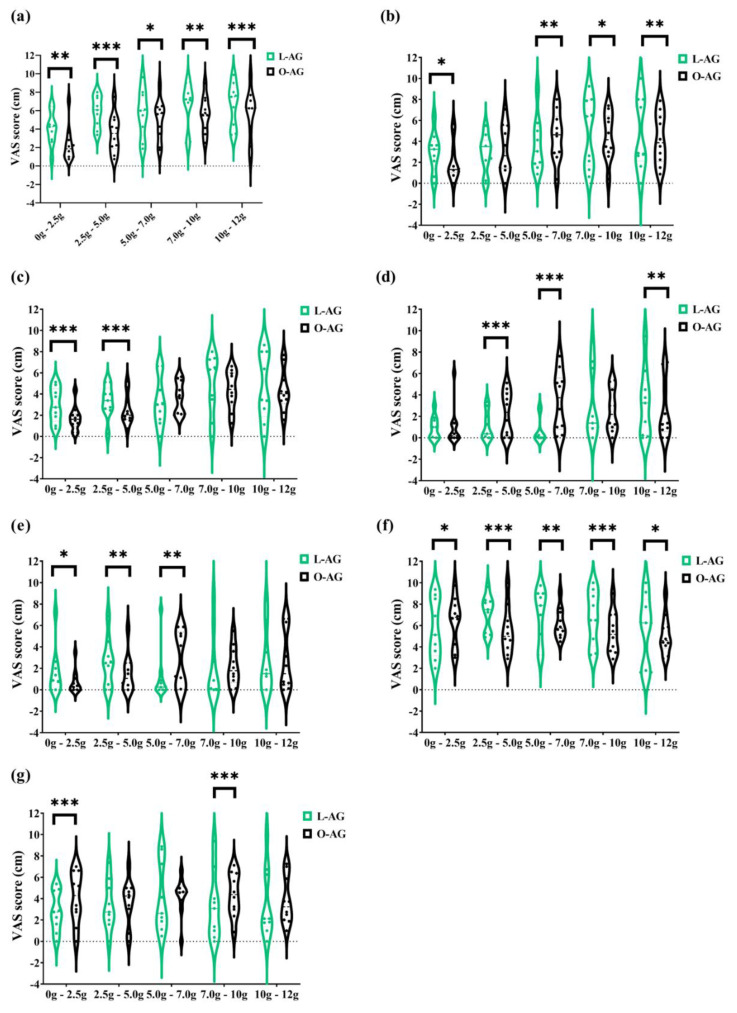
The contrast of gastrointestinal adaptation and subjective satiety and appetite ratings between lean and obese individuals after 2.5, 5, 7, 10, and 12 g/day of agavins during a five-week dose-escalation intervention. Violin plots with individual records are presented. (**a**) Flatulence; (**b**) Bloating; (**c**) Borborygmi; (**d**) Abdominal pain; (**e**) Diarrhea; (**f**) Satiety; (**g**) Appetite. ** p* < 0.05, *** p* < 0.01, **** p* < 0.001 inter-group differences for the same dose (unpaired *t*-test with Welch’s correction). L-AG, lean group consuming agavins; O-AG, obese group consuming agavins.

**Figure 5 foods-11-00670-f005:**
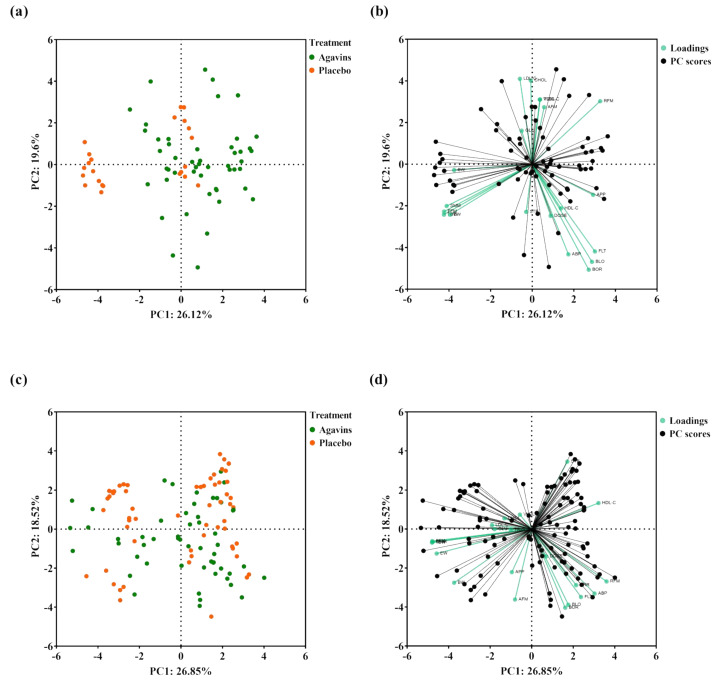
Principal component analysis (PCA) of variables measured in this study. (**a**) PCA scores plot of lean-agavins and lean-placebo groups; (**b**) Biplot of the lean-agavins group; (**c**) PCA scores plot of obese-agavins and obese-placebo groups; (**d**) Biplot of the obese-agavins group. DOSE, agavins/placebo daily dose; FLT, flatulence; BLO, bloating; BOR, borborygmi; ABP, abdominal pain; DRR, diarrhea; GLU, glucose; CHOL, total cholesterol; TRIG, triglycerides; HDL-C, high-density lipoprotein cholesterol; LDL-C, low-density lipoprotein cholesterol; VLDL-C, very-low-density lipoprotein cholesterol; BW, body weight; RFM, relative fat mass; AFM, absolute fat mass; FFM, fat-free mass; SMM, skeletal muscle mass; TBW, total body water value; EW, extracellular water value; STE, subjective satiety; APP, subjective appetite.

**Table 1 foods-11-00670-t001:** Baseline characteristics of lean and obese participants enrolled to evaluate and compare five gastrointestinal symptoms in response to increasing doses of agavins for a five-week period.

	Characteristics	Lean (BMI ^a^ 18.5–24.9 kg/m^2^)	Obese (BMI ≥ 30 kg/m^2^)
**Demographics**	*n*	16	22
	Male, *n* (%)	8 (50)	6 (27)
	Female, *n* (%)	8 (50)	16 (73)
	Age (years)	38. 8 ± 10.4	41.4 ± 8.9
**Clinical variables**	BMI (kg/m^2^)	23.7 ± 1.3	33.01 ± 3.5
	Bodyweight (kg)	63.19 ± 8.1	86.96 ± 12.7
	Relative fat mass (kg)	29.47 ± 5.5	42.26 ± 5.8
	Absolute fat mass (kg)	18.50 ± 2.7	36.79 ± 7.8
	Fat-free mass (kg)	44.96 ± 8.1	50.17 ± 8.7
	Skeletal muscle mass (kg)	20.32 ± 4.3	23.04 ± 5.1
	Total body water (L)	32.9 ± 5.7	37.15 ± 6.4
	Extracellular water (L)	13.69 ± 2.2	16.42 ± 2.4
	Fasting glucose (mg/dL)	83.06 ± 7.8	84.45 ± 11.7
	Triglycerides (mg/dL)	134.6 ± 64.1	158.1 ± 84.1
	CHOL-T ^b^ (mg/dL)	201.6 ± 38.6	175.1 ± 33.1
	HDL-C ^c^ (mg/dL)	51.17 ± 11.2	41.36 ± 10.2
	LDL-C ^d^ (mg/dL)	123.8 ± 31.2	102.1 ± 22.6
	VLDL-C ^e^ (mg/dL)	26.96 ± 12.8	31.64 ± 16.8

Data are presented as mean ± standard deviation. BMI ^a^, body mass index; CHOL-T ^b^, total cholesterol; HDL-C ^c^, high-density lipoprotein cholesterol; LDL-C ^d^, low-density lipoprotein cholesterol; VLDL-C ^e^, very-low-density lipoprotein cholesterol.

**Table 2 foods-11-00670-t002:** Dietary intake in lean and obese participants randomized to agavins or placebo groups.

	L-AG ^a^	L-PL ^b^	O-AG ^c^	O-PL ^d^
Total energy (kcal)	1367 ± 127.4	1286.2 ± 69.9	1938 ± 217.2	1623.5 ± 175.8
Carbohydrates (%)	44.7 ± 3.9	43 ± 1.9	47.1 ± 3.3	51.4 ± 4.5
Proteins (%)	20.3 ± 1.2	23 ± 2.7	17.1 ± 1.4	18.8 ± 1.9
Fat (%)	35 ± 3.9	34.1 ± 2.7	35.9 ± 3.3	29.8 ± 3.3

L-AG ^a^, lean-agavins; L-PL ^b^, lean-placebo; O-AG ^c^, obese-agavins; O-PL ^d^, obese-placebo. No significant differences (*p* < 0.05) between agavins and placebo groups at baseline, unpaired *t*-test with Welch’s correction.

## Data Availability

The data are not publicly available due to privacy.

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
