# Peer review of "Agavins Impact on Gastrointestinal Tolerability-Related Symptoms during a Five-Week Dose-Escalation Intervention in Lean and Obese Mexican Adults: Exploratory Randomized Clinical Trial"

_foods, 2022, doi:10.3390/foods11050670_

Round 1

Reviewer 1 Report

This is an excellent study . It is well designed, and results are a promising initial study on the relevance of Agavins as a probiotic. 

The discussion is excellent, and authors have addressed the limitation of the study.

Some areas of Introduction and Discussion can be reduced with better focus on the many functional benefits of Agavins. 

Another minor weakness that details of chemistry of Agavins and consistency of dose is missing. 

Author Response

We, the authors, deeply appreciate all your comments and are sure they will improve this manuscript. We detail herein all changes included in the latest version of our submitted manuscript (in blue) to fulfilled referees’ suggestions and comments:

  1. Line 42: addition of a reference to sedentary lifestyle as a contributing factor to the increasing prevalence of obesity
  2. Line 58: precisions on agavins chemistry
  3. Line 69: addition of “agavins” to “5 g/day of agavins”
  4. Line 75: aim of the study was rewritten
  5. Line 130: inclusion of a justification for agavins’ doses
  6. Line 211: blank space suppressed

Cordially,

Mercedes G. López

Reviewer 2 Report

In this study you  provided information on some symptoms that may cause GI distress when agavins are consumed and even more with increased daily doses up until to 12 g/day targeting treated groups or healthy controls. This study gave more information but not clear answer. As you mentioned in the future study you should form larger test groups and design longer and larger randomized clinical trial for the definite answers.

for line 43: physical activity should be also added with some reference for it

line 68: it should be: 5g/day of agavins

lines 74-77: in this part, the aim of this study should be better and more clear described and stated

Author Response

(The authors gave the same response as above.)
